# Quantum plasmonics pushes chiral sensing limit to single molecules: a paradigm for chiral biodetections

Chi Zhang[1,7], Huatian Hu [2,3,7], Chunmiao Ma[4,7], Yawen Li[1], Xujie Wang[1], Dongyao Li[4], Artur Movsesyan [5,6], Zhiming Wang [5], Alexander Govorov [6], Quan Gan [4] ✉ & Tao Ding [1] ✉

Chiral sensing of single molecules is vital for the understanding of chirality and their applications in biomedicine. However, current technologies face severe limitations in achieving single-molecule sensitivity. Here we overcome these limitations by designing a tunable chiral supramolecular plasmonic system made of helical oligoamide sequences (OS) and nanoparticle-on-mirror (NPoM) resonator, which works across the classical and quantum regimes. Our design enhances the chiral sensitivity in the quantum tunnelling regime despite of the reduced local E-field, which is due to the strong Coulomb interactions between the chiral OSs and the achiral NPoMs and the additional enhancement from tunnelling electrons. A minimum of four molecules per single-Au particle can be detected, which allows for the detection of an enantiomeric excess within a monolayer, manifesting great potential for the chiral sensing of single molecules.

Chiral molecule sensing has significant implications for the drug industry and biomedical applications[1,2]. However, chiroptic detection on a single-molecule level still remains elusive for chiral molecules with traditional single-molecule techniques such as single-molecule Raman spectroscopy[3], fluorescence spectroscopy[4], and nanopore sensing technology[5], which show poor capability in discriminating the enantiomers. The development of nanophotonics has provided a promising approach towards this end with the assistance of superchiral fields and plasmon-coupled circular dichroism (PCCD)[6,7], which can be characterized with single-particle-based circular differential scattering (CDS) spectroscopy[8]. The synergy of different enhancement mechanisms has dramatically improved the sensing capability down to a few biomolecules[9–11]. However, due to the limited control on the gap size

and orientation of the biomolecules therein[12], the CDS enhancement (CDSE) is small[13] and the chiral sensing limit is still above the level of single chiral molecules[14].

Extreme nanophotonic construct made of Au nanoparticle-on-mirror (NPoM) with sub-nm gap has shown huge local E-field enhancement up to ~1000, which exhibits vibrant functionality on single-molecule strong coupling, active quantum plasmonics, and optomechanics[15]. This NPoM system can be easily fabricated via the drop-casting method, providing single-particle-based nanocavities with out-of-plane electric field dipole. Thus, it can serve as a sensitive platform for single chiral molecules where the quantum tunneling effect may perturb the chiroptic responsivity[16]. This quantum tunneling effect is commonly recognized to be detrimental to the plasmonic

[1]Key Laboratory of Artificial Micro/Nano Structure of Ministry of Education, School of Physics and Technology, Wuhan University, 430072 Wuhan, China. [2]Hubei Key Laboratory of Optical Information and Pattern Recognition, Wuhan Institute of Technology, 430205 Wuhan, China. [3]Center for Biomolecular Nanotechnologies, Istituto Italiano di Tecnologia, Via Barsanti 14, Arnesano, LE 73010, Italy. [4]School of Chemistry and Chemical Engineering, Huazhong University of Science and Technology, 430074 Wuhan, China. [5]Institute of Fundamental and Frontier Sciences, University of Electronic Science and Technology of China, 610054 Chengdu, China. [6]Department of Physics and Astronomy, Ohio University, Athens, OH 45701, USA. [7]These authors contributed equally: Chi Zhang, Huatian Hu, Chunmiao Ma. ✉e-mail: ganquan@hust.edu.cn; t.ding@whu.edu.cn

enhancement due to the reduced local E-field in the gaps. However, this might not always be true for the optical chirality of a tunneling structure, which is ambiguously understood.

Here we adopt oligoamide sequences (OS) composed of quinoline-based octamer and a stereogenic center as the gap medium, which can fold into helices with full P or M-helicity control (see Supplementary Information for detailed synthesis and characterizations, Supplementary Figs. 1–30). These helical superstructures can be further duplexed into enantiomers of double helices (DHs) via π-π interactions (Fig. 1a), which show distinctive absorption and circular dichroism (CD) in the ultraviolet (UV) region, while their racemic mixture shows no CD response across the whole spectrum (Fig. 1b, c). When these racemic OS are placed in the ultrathin metal cavity of NPoM (NPoM/OS), strong local chirality can be resolved with huge CDSE (up to ~6 × 10⁶) despite of the quantum tunneling effect, which is significantly larger than other nanophotonic structures such as nanoislands[17], nanorods (dimers)[12,13], and nanochains[18] (Fig. 1d). We reveal additional Coulomb interactions between the chiral supramolecules and tunneling electrons compensate the decrease of local E-field incurred by tunneling, which induces dramatic increase of CDSE in the quantum regime. This generalized model of plasmon-enhanced chirality not only reveals the full picture of light-chiral matter interaction at different scales but also serves as a basic guideline for the design of advanced sensing platforms for chiral molecules in general.

## Results
### Chiral sensing based on NPoMs
We make a self-assembled monolayer (SAM) of these well-defined chiral helices on Au films, followed by drop-casting Au NPs on top to form the NPoM/OS system. Due to the ultrathin spacer of the SAM (< ~2 nm), the resulting Au NPoM exhibits intense electric field confinement in the nanogap (inset of Fig. 2a), whose intensity increases exponentially with the decrease of gap size (blue dash line in Supplementary Fig. 31). As the chiral dipole of the helices aligns to the out-of-plane dipole of the nanogaps, the chiroptic response of this supramolecular NPoM system is significantly enhanced due to the synergy of E-field enhanced CD and PCCD ($CD_{tot} = CD_{mol} + CD_p$)[19]. Besides, the field enhancement of the NPoM structure leads to strong optical chirality ($C/|C_0|$) to the chiral light with intensity up to 50 (Fig. 2b and see Methods for the calculation details). The superchiral field ($C/|C_0|$) of this NPoM system increases as the gap size decreases even in the quantum regime (blue dash line in Supplementary Fig. 32), suggesting large chiroptic enhancement of this plasmonic system. The chiroptic response of the Au NPoM/P-helix and M-helix hybrid structures is robustly observed in CDS spectra over 50 individual NPs, most of which show CDS intensity at −20% and +20% for NPoM/P-helix and NPoM/M-helix, respectively (Fig. 2c). These CDS spectra are mainly from the PCCD of the chiral molecules as negligible CDS response was observed for Au NPoMs with achiral molecules (1,4-Benzenedithiol) (Supplementary Fig. 33). The variation of the CDS intensity is mainly due to the differences in the size/shape of Au NPs and the number of helical supramolecules in the gap (Supplementary Fig. 34). The substrate effect may also contribute to such variability. For better comparison, we choose four different-sized Au NPoMs with P- and M-helices, which present different scattering responses to the incidences of left and right circular polarization (LCP and RCP) (Fig. 2d–i). Careful analysis of these chiral scattering reveals that different-sized Au NPoMs indeed show shifted plasmonic scattering (Fig. 2d, e, g, h),

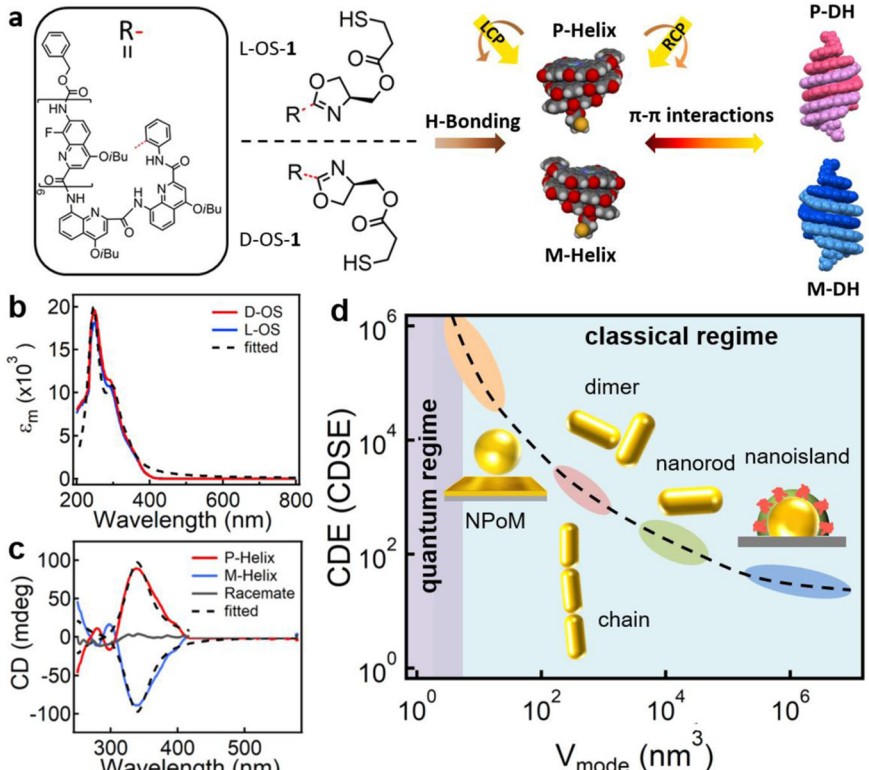

**Fig. 1 | Concept of chiral biosensing based on extreme nanophotonics.**
**a** Schematic of the enantiomers of OS molecules and their chiral transfer process, forming P-/M-Helices via H-bonding and P-/M-DHs via π-π interactions. The difference in the absorption of LCP and RCP light generates the CD spectra. **b** UV–Vis absorption and **c** CD spectra of P-/M-helix and their racemate. The dash lines are fitted spectra with parameters indicated in the simulation section. **d** Change of CD enhancement (CDE) or CDSE with decreasing mode volume of plasmonic nanostructures. Insets are schematics of several typical plasmonic nanostructures, from left to right are NPoM, chain, dimer, nanorod and nanoisland. The colors of light orange, pink, green, and blue represent the typical correlation between mode volume and CDE (CDSE) value of NPoM, chain/dimer, nanorod, and nanoisland, respectively. The dashed line serves as a guide for eyes only.

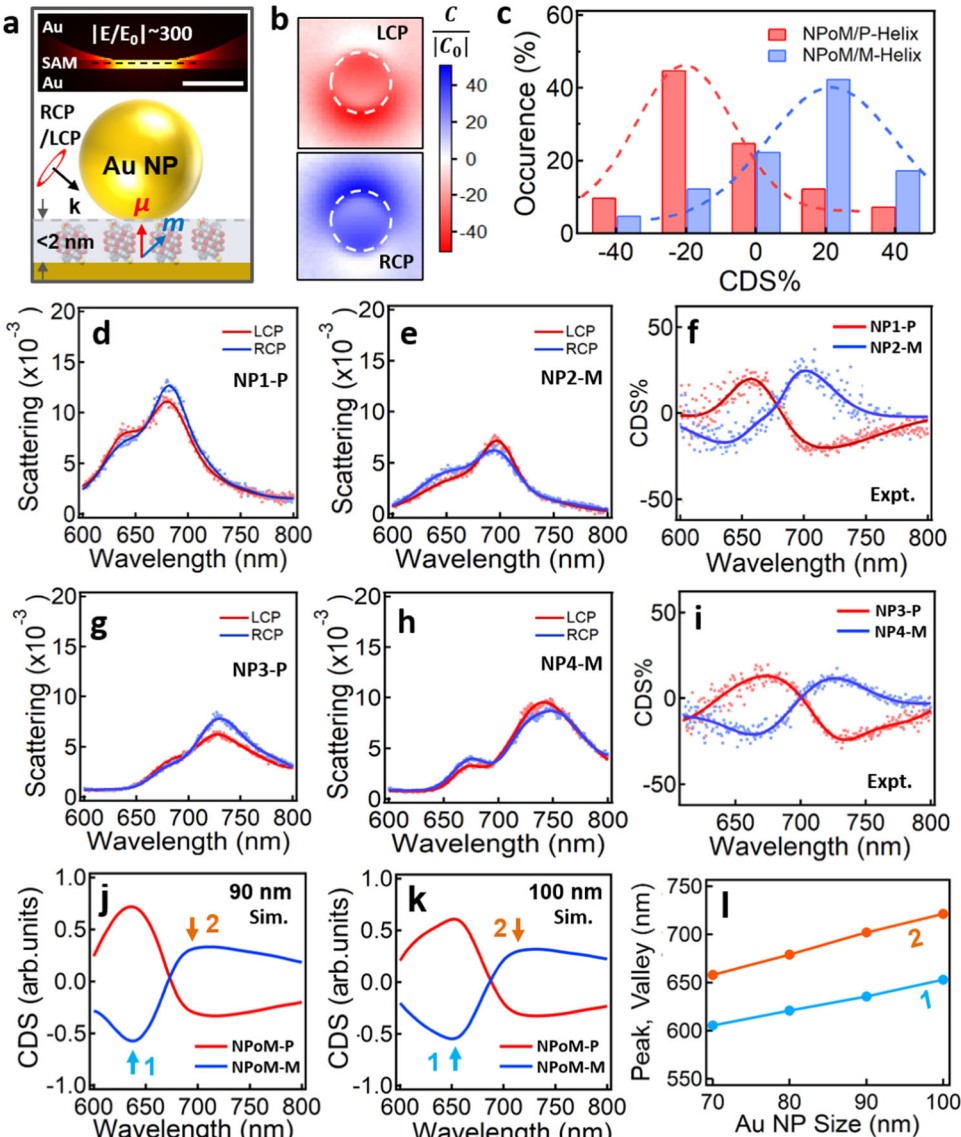

**Fig. 2 | NPoM-assisted chiral sensing of a few chiral helices. a** Schematic of Au NPoM nanostructure with helical SAM sandwiched in the nanogap. Inset is the electric near-field profile (at a wavelength of 730 nm) of Au NPoM with a SAM (thickness: 1.3 nm) in the nanocavity. RCP: right circular polarization, LCP: circular polarization. **b** Optical chirality ($C/|C_0|$) mapping in the nanocavity (black dash line marked in the inset of **a**) with achiral SAM under LCP and RCP light illumination (730 nm). **c** Statistics of the CDS intensity measured over ~50 Au NPoMs with P-/M-Helices. Scattering spectra of Au NPoM with **d, g** P-helix and **e, h** M-helix in the nanogaps under LCP and RCP dark field incidences, and **f, i** their corresponding CDS spectra. **j, k** Simulated CDS spectra of Au NPoM/OS with Au NP size of **j** 90 nm and **k** 100 nm. 1, 2 refer to the valley and peak position. **l** Change of CDS peak/valley with the change of Au NP size.

which results in shifted CDS response (Fig. 2f, i). Finite element method (FEM) is employed to numerically simulate the CDS spectra of different-sized Au NPoMs (Fig. 2j, k and see Methods for simulation details), which agrees well with the experimental results (Fig. 2f, i). The peaks/valleys of the CDS spectra redshift as the size of Au NPs increases (Fig. 1l) but their intensity appears almost the same (Supplementary Fig. 35).

The packing density of DH in the SAM film is ~0.2 nm$^{-2}$ as measured by quartz crystal microbalance (Supplementary Fig. 36a, b), suggesting ~16 DHs with a facet of 10 nm and ~4 DHs with a facet of 5 nm (Supplementary Fig. 36c). For 80 nm Au NPoM, the facet size normally ranges from 5 to 10 nm[20], thus around 10 DHs are expected to contribute the CDS intensity measured with ~2% per DH, which can be reasonably detected by this NPoM system. Thus, it holds great potential for chiral single-molecule detection.

## Revealing the local chirality of racemates using Au NPoMs

Another characteristic of single chiral molecule sensitivity is that the racemic mixture that appears achiral as an entity can be locally resolved as enantiomer excess (ee) due to the population fluctuation in the single Au NPoM gap. Clearly, in this NPoM configuration with facets of 5–10 nm and DH numbers of 5–15, a high probability of local ee could statistically appear (see Methods for detailed calculation), which renders a diverse chiroptical response with the NPoM system. Indeed, we found that the racemic mixture with no CD response (Supplementary Fig. 37) can either exhibit CDS response of opposite signs (Fig. 3a, b and Supplementary Fig. 38a, b) or negligible CDS response (Fig. 3c and Supplementary Fig. 38c). And a significant portion (47%) of Au NPoMs appears to show CDS intensity of ~20%, although 42% of NPoMs shows no CDS response (Fig. 3d). The detailed statistics of the CDS occurrence appears Gaussian-like distribution which matches exactly the probability of ee based on 10 DH

molecules in the monolayer (Fig. 3e and see Methods for the detailed calculation).

## Contribution of quantum tunneling to the CDSE of Au NPoM

The chiral sensing capability of Au NPoM can be further improved with reduced gap size as predicted by the enhanced E-field and dissymmetric factor (Supplementary Figs. 31 and 32). Here we devise a set of OS DHs of different sizes (Fig. 4a and Supplementary Table 1 for the detailed comparison), which can be dissociated via heating to further reduce the gap size (Fig. 4b)[21]. As the size of the gap decreases with OS molecules of different states, the plasmon resonances of the Au NPoM first redshifts from 695 to 738 nm and then blue-shifts back to 726 nm due to the quantum tunneling (gray dash line in Fig. 4c and Supplementary Fig. 39)[22]. Clearly, these quantum plasmonics cannot be analytically predicted via the circuit model (gray solid line in Fig. 4c), which normally applies to the classical regime (CR)[23]. With quantum-corrected model (QCM)[22], the scattering spectra of Au NPoMs/OS can be numerically simulated from classical to quantum regime (QR) (see mapping in Fig. 4c and Methods for the simulation details), which agree well with the experimental results (gray dash line in Fig. 4c). The CDS spectra of the corresponding Au NPoM/OS system are collected over 30 individual NPs and statistically plotted in Fig. 4d. As expected by classical model (CM), the absolute maximum CDS intensity increases with the decrease of gap size in the CR but decreases at the quantum transition, which then increases again in the QR (red dash line in Fig. 4e). This trend agrees pretty well with the calculated optical chirality (red dash line in Supplementary Fig. 34) and CDSE (green dash line in Fig. 4e and Supplementary Fig. 40) based on QCM, suggesting the electron tunneling induces the decrease of CDS intensity at the classical/quantum transition ( ~ 0.7 nm) due to the reduced intensity of E-field (blue dash line in Supplementary Fig. 31). However, additional contribution of Coulomb interactions between chiral OS and tunneling electrons further enhances the CDS intensity below the quantum tunneling limit (~0.7 nm). Thus, a third term of absorption rate induced by the chiral tunneling electrons ($Q_t$) should be introduced in the expression of full chiral light-matter interaction as

$Q_{tot} = Q_{mol} + Q_p + Q_t$. Here,

$$Q_t = \int_0^{V_g} dV \mathbf{J_t} \cdot \mathbf{E} \tag{1}$$

where $\mathbf{E}$ is the local E-field in the gap, $\mathbf{J_t}$ is the tunneling current and $V_g$ is the volume of the nanogap. The overall trend of enhancement is not disturbed by the orientation of the molecules in the gap although the alignment of the molecular dipole to the plasmonic dipole can further increase the CDSE by 3-fold (Supplementary Fig. 41).

## Discussion

In summary, the high sensitivity of NPoM on the chiral helices is supported by the giant Columbic interactions of electron tunneling along with the PCCD mechanism. The benefits of this chiral NPoM system compared to other chiral nanophotonic structures are enormous (Table 1). First, it can be facilely and robustly fabricated with small mode volume (~37 nm³), which strongly boosts the chiral light-matter interactions with intense local E-field (~300), while previous chiral sensing systems are mostly based on large biomolecules such as proteins and DNA, which cannot achieve such large enhancement in CDS. Second, it presents a large g-factor based on single-particle CDS spectra, with the potential capability for detecting single chiral molecules, outperforming many previous reports[6,7,11]. Last but not least, the unique design of the chiral supramolecules endows the accessibility of quantum plasmonic regime with the alignment of chiral dipoles to the plasmon dipole, which reveals additional CDSE from the quantum tunneling effect, unveiling the full description of chiral light-matter interactions. The design principles and measurement techniques revealed in this article can serve as a guideline for nanosensors with superior performances of enantiomer discrimination, biomedical diagnosis, and drug quality control.

## Methods

### Synthesis of OSs and characterizations

All chemicals and solvents were purchased from commercial suppliers and were used without further purification unless specified. For

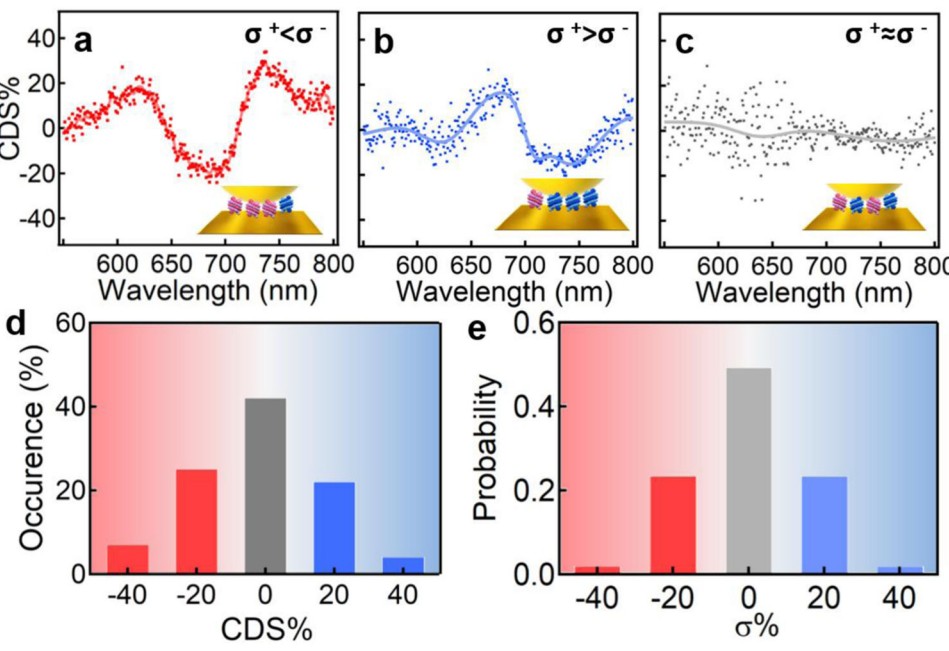

**Fig. 3 | Local chirality of racemates revealed by Au NPoMs. a–c** CDS spectra of Au NPoMs with racemate of OS DHs in the gaps. Schematic insets in **a–c** represent the occasions that the number of P-DHs is **a** more than, **b** less than, and **c** equal to that of M-DHs, respectively. **d** Statistical occurrence of the CDS intensity measured based on ~50 Au NPoMs/DHs. **e** Probability of enantiomer excess based on 10 DHs in the nanogaps of Au NPoM.

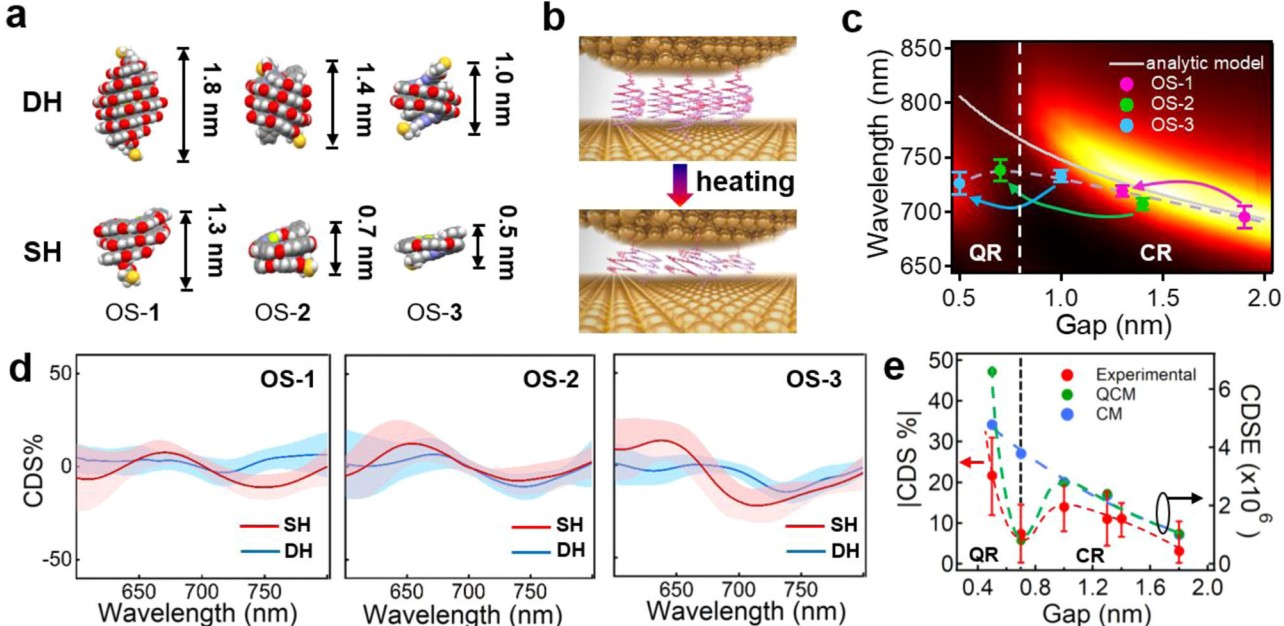

**Fig. 4 | Contribution of quantum tunneling to the CDSE of Au NPoM/OS system.**
**a** Configurations of double and single helices of OS-1, OS-2, and OS-3. The sizes are based on modeling by SPARTAN. **b** Schematic of Au NPoM/OS before and after heating, showing decreased gap size. **c** Change of plasmon resonances with gap size. The background mapping is the simulated scattering spectra of Au NPoM at different gap sizes based on QCM. The pink, green, and blue arrows represent the experimental data of plasmon resonance change with gap size after heating OS-1, OS-2 and OS-3, respectively. The data points are averaged and statistically plotted over 30 Au NPoMs and the gray dash line serves as the guide for eyes only. The gray

sold line is the calculated change of plasmon resonance with gap size based on the circuit model. The error bars are the standard deviation of plasmon peaks.
**d** Averaged CDS spectra of Au NPoM/OS in the state of DH and SH. The light blue and pink bands represent the standard deviation of 10 CDS spectra in each case.
**e** Change of the absolute CDS intensity (dips around 750 nm) extracted from **d** (red dash line) and calculated CDSE (green and blue dash lines) with gap size. The green and blue dots represent the CDSE calculated with QCM and CM, respectively. The black dash line highlights the transition from CR to QR. The error bars are the standard deviation of CDS intensities at different gap sizes.

detailed synthesis, see supplementary text in the next section. Dichloromethane (DCM) and diisopropylethylamine (DIEA) was distilled over $CaH_2$ prior to use. Column chromatography was carried out on Merck GEDURAN Si60 (40–63 μm).

NMR spectra were recorded on Bruker AVANCE 400 (400 MHz) spectrometers. Chemical shifts were calibrated by $CDCl_3$ (7.26 ppm for $^1H$ NMR, 77.16 ppm for $^{13}C$ NMR) and by DMSO-$d_6$ (2.50 ppm for $^1H$ NMR, 39.52 ppm for $^{13}C$ NMR). All chemical shifts ($\delta$) are quoted in ppm and coupling constants ($J$) are expressed in Hertz (Hz). The following abbreviations are used for convenience in reporting the multiplicity for NMR resonances: s = singlet, d = doublet, t = triplet, and m = multiplet. Data processing was performed with Topspin 2.0 software.

High-resolution electrospray ionization mass spectrometry (ESI-MS) was performed on a micro TOF II instrument featuring a Z spray source with electrospray ionization. All the molecular structural characterizations and summary are shown in Supplementary Figs. 1 to 30 and Supplementary Table 1.

### Sample preparation

A SAM of double helices was first formed by immersing the Au films (70 nm) in the chloroform solution of OS-1 (10 mM) for 12 h, followed by rinsing with pure chloroform. 5 μL of Au NP dispersion (Nanopartz) of varied sizes was drop casted on the SAM for 20 min, which was then blow-dried using a nitrogen gun. For temperature-induced dissociation, the samples were immersed in hot chloroform (60 °C) for 10 min to achieve reduced gap size with single helices. All the samples were vacuumed for 30 min to dry thoroughly before taking dark field (DF) scattering spectra.

### Characterizations

CD spectra of OSs are measured with Chirascan (Applied Photophysics Ltd) at a concentration of 0.1 mM. The packing density of the SAM was

characterized by quartz crystal microbalance (QSense E1, Boilin Scientific), and the morphology of the Au NPoM was captured with SEM at an accelerating voltage of 5 kV. DF scattering spectra of each individual Au NPoM were recorded using a customized DF microscope (BX53, Olympus) through 100x objective (NA = 0.9) with a 50 μm optical fiber coupled to a spectrometer (QEPro, Ocean Optics). To acquire CDS spectra, RCP and LCP incidences were generated by passing halogen light through a linear polarizer and a quarter-wave plate with fast axial alignment of ±45° to the polarization, which was then focused on the Au NPoM through the ×100 objective. We have compared the CDS spectra obtained with symmetric and asymmetric incidences and find the latter shows stronger intensity although the lineshape looks the same (Supplementary Fig. 42). While for symmetric incidence, the same distribution span from negative to positive was observed but with much weaker intensity (mostly reduced to zero) (Supplementary Fig. 43). Thus, for the optimum signal-to-noise ratio of CDS spectra, we adopt the asymmetric incidence for the CDS measurement in the report unless stated otherwise. The scattering spectra of both incidences ($S_{LCP}$ and $S_{RCP}$) were collected as normal DF scattering and the CDS spectra can be obtained via the formula

$$CDS\% = \frac{2(S_{LCP} - S_{RCP})}{(S_{LCP} + S_{RCP})} \times 100\% \qquad (2)$$

This CDS value is equivalent to g-factor. The DF and CDS spectra over 30 randomly selected Au NPoMs were measured and statistically plotted.

### Simulations

**CDS calculation.** The finite element method was applied to simulate the electric near field, scattering spectra, and CDS spectra of the Au NPoM. The geometry contains an 80 nm AuNP with a 6 nm facet on Au

**Table 1 | Comparison of relevant nanophotonic structures used for chiral sensing**

| Geometry | Method | $V_{mode}$/(nm$^3$) | Detection limit/(M) | g-factor | References |
|---|---|---|---|---|---|
| Nanoparticles | Chemical synthesis | ~$4 \times 10^5$ | ~$10^{-12}$–$10^{-6}$ | ~0.2–0.4 | 33,34 |
| Dimers Nanorods /Chains | Self-assembly | ~$10^3$–$10^4$ | ~$10^{-18}$–$10^{-21}$ | ~$10^{-2}$–$10^{-3}$ | 12,35,36 |
| Nanocuvettes Islands Cylinders Shurikens Gammadia | Lithography | ~$10^5$ | ~a few to $10^{-21}$ | ~$10^{-2}$–$10^{-1}$ | 14,17,28,37–39 |
| NPoM | Drop-cast | ~37 | Single molecules | 0.2–0.3 | This work |

film spaced by chiral molecular films with different thicknesses. An oblique plane wave incidence (64° corresponding to 0.9NA) with circular polarization was applied to excite the system. The permittivity of Au was adopted following Johnson-Christy et al.[24], while the chiral molecules in the nanogap follow unique constitutive equations which were modified in COMSOL[13,25]:

$$\mathbf{D} = \varepsilon_0 \varepsilon \mathbf{E} - \frac{i\kappa}{c} \mathbf{H} \tag{3}$$

$$\mathbf{B} = \mu_0 \mu \mathbf{H} + \frac{i\kappa}{c} \mathbf{E} \tag{4}$$

where $\varepsilon_0 \varepsilon$ and $\mu_0 \mu$ are the permittivity and permeability of the chiral medium, respectively. $\kappa$ represents the chiral strength of the medium. The parameters were modeled as,

$$\varepsilon = \varepsilon_b - \sum_{j=1}^{n} \gamma_j \left( \frac{1}{\hbar\omega - \hbar\omega_{0,j} + i\Gamma_j} - \frac{1}{\hbar\omega + \hbar\omega_{0,j} + i\Gamma_j} \right) \tag{5}$$

$$\kappa = \sum_{j=1}^{n} \beta_j \left( \frac{1}{\hbar\omega - \hbar\omega_{0,j} + i\Gamma_j} + \frac{1}{\hbar\omega + \hbar\omega_{0,j} + i\Gamma_j} \right) \tag{6}$$

$$\mu = 1$$

where $\varepsilon_b = 1.69$ is the dielectric constant of the background and the coefficients $\gamma_j$ and $\beta_j$ are the amplitude of absorption and chirality, respectively. $\omega_{0,j} = 2\pi c / \lambda_{0,j}$, where $\lambda_{0,j}$ is the absorbance peak wavelength of the chiral molecules with linewidth of $\Gamma_j$. Here, all the parameters were fitted by multiple Lorentzian functions to the experimental spectra.

Specifically, the linewidths $\Gamma$ and absorption amplitude $j$ can be determined by the molar extinction coefficient[13,26] with the analytical formula,

$$\varepsilon_{ext} = \sum_{j=1}^{n} \frac{\gamma_j}{n \cdot n_0 \cdot c_0} \frac{\omega \cdot N_A \cdot 10^{-4}}{0.23} \left( \frac{\Gamma}{(\hbar\omega - \hbar\omega_0)^2 + \Gamma^2} - \frac{\Gamma}{(\hbar\omega + \hbar\omega_0)^2 + \Gamma^2} \right) \tag{7}$$

Here $n$ represents the background refractive index, $c_0$ is the vacuum light speed, $N_A$ is the Avogadro constant, $n_0$ is the effective density of dipole pairs of OS molecules $(0.7\,\text{nm})^{-3}$. By fitting the experimental extinction spectra of the chiral molecules (Fig. 1b), we could obtain the $\Gamma_1 = 0.3977$ eV, $\Gamma_2 = 0.3619$ eV, $\Gamma_3 = 0.25$ eV; $\gamma_1 = 0.4299$ eV, $\gamma_2 = 0.2053$ eV, $\gamma_3 = 0.02401$ eV.

For chirality, we adopt the analytical solution from ref. 27, where a thin chiral molecule film has a CD response with analytical expression as

$$\text{CD} = -\tan^{-1} \left[ \tanh\left( 2k_0 w_i \text{Im}\{\kappa\} \right) \right] \tag{8}$$

Given that the $k_0$ is the vacuum wave vector and $w_i$ is the thickness of the film (~1 mm), we can obtain the chirality amplitude $\kappa$

by fitting out its Lorentzian parameters (dash lines in Fig. 1c) as following: $\beta_1 = 3.387 \times 10^{-9}$ eV, $\beta_2 = 2.217 \times 10^{-9}$ eV, $\beta_3 = -1.222 \times 10^{-8}$ eV; $\omega_{0,1} = 5.03$ eV, $\omega_{0,2} = 4.115$ eV, $\omega_{0,3} = 3.65$ eV. Similar to ref. 28, we also artificially magnified the chirality parameter $\kappa$ by a factor of $2.5 \times 10^7$ to ensure reasonable computational accuracy and time. This change does not affect the lineshape of the spectra and its effect can be eliminated by further normalization.

**CDS enhancement (CDSE).** CDSE is obtained by normalizing the CDS spectra to the one without NPoM structures as CDSE = $|\text{CDS}_{\text{NPoM/OS}}|/|\text{CDS}_{\text{OS}}|$. The normalization factor (denominator) indicates the intrinsic CDS contributed solely from a collection of molecules with a volume overlapped with the NPoM mode. In simulation, the $\text{CDS}_{\text{OS}}$ is the difference of the scattering by LCP and RCP excitation, from a molecular "disk" without NPoM. The height $H$ of the cylinder is the thickness of the molecule layer (e.g., 2.0 nm, 1.8 nm, 1.5 nm etc.), while the diameter $2R$ of the cylinder is determined by the mode volume ($V_m$) of the NPoM, $R = \sqrt[3]{\frac{3}{4} V_m / \pi}$ (see the calculation of mode volume below). Since the NPoM structure has the most confined field in the nanogap under the bottom facet, this estimation is reasonable. The orientation of the molecules in the nanogap could also be considered with the formalisms derived in reference[29]. Specifically, we assumed a vertical electric dipole with a magnetic dipole of 20 degrees for the supramolecules in the gap and calculated the CDSE based on the formula mentioned above (Supplementary Fig. 41).

**Quantum-corrected model (QCM).** QCM was applied when analyzing the NPoM with sub-nm gaps in the quantum regimes where quantum tunneling could arise. The permittivity of the gap material is phenomenologically modified by a relation[20,30]

$$\varepsilon_g(l, \omega) = \varepsilon_\infty - \frac{\omega_g^2}{\omega \left( \omega + i\gamma_g e^{(l/l_c - \alpha)} \right)} \tag{9}$$

Here, $l$ is the thickness of the tunneling area, $\varepsilon_\infty = 1.96$, $\omega_g = 1.3597 \times 10^{16}$ rad/s, $\gamma_g = 1.0483 \times 10^{14}$ rad/s, $l_c = 0.11$ nm, $\alpha = 0.8$. The morphology of the quantum tunneling junction could be approximately considered as a small disk whose cross-section is equivalent to the facet area (6 nm in diameter) with the height of OS molecules. In this way, we can calculate the scattering spectra of the NPoM with different gap thicknesses (Fig. 3c), which match the experimental results.

**Mode volume.** The mode volume of a plasmonic nanostructure defines the fineness of the confinement of the plasmonic field. It can be calculated from the Purcell factor $F_p$ together with the quality factor $Q$, excited by a dipole situated at the maximum field position in the nanogap[31], according to the well-known Purcell formula

$$F_p = \frac{3}{4\pi^2} \left( \frac{\lambda_0}{n} \right)^3 \frac{Q}{V} \tag{10}$$

Here, $\lambda_0$ is the vacuum wavelength and $n$ is the refractive index of the material in the cavity.

In Table 1, the mode volumes of each characteristic structure are calculated with the parameters as follows: (a) 120 nm gold nanocube of 20 nm rounding curvatures on corners and sides; (b) 100 nm × 40 nm nanorod dimer with side-by-side and end-to-end configurations with a gap of 1 nm; (c) nanoshuriken nanostructure with lateral dimensions of 500 nm and height of 80 nm on a silicon substrate; (d) 80 nm Au NPoM with a gap of 1 nm and facet of 6 nm. Note that the position of the dipoles is at the maximum field which efficiently excites the resonances for different structures (e.g., (a) at the corner of the nanocube, (b) in the gap of the nanorod dimer, (c) at the ends of shuriken arms, (d) at the center of the NPoM cavity).

**Optical chirality.** Optical chirality $C$ was calculated by[32],

$$C = -\frac{\omega \varepsilon_0}{2} \mathrm{Im}\left(\mathbf{E}^* \cdot \mathbf{B}\right) \tag{11}$$

where $\mathbf{E}$ and $\mathbf{B}$ are the near-field electric and magnetic fields, $\omega$ is the angular frequency, $\varepsilon_0$ is the vacuum permittivity. Both classical and quantum-corrected models are applied to calculate the $C$ of NPoM at different gap sizes, whose maximum intensities are plotted in Supplementary Fig. 32.

**Probability of enantiomer excess with a few chiral OS molecules in the gap.** For the calculation of local chiral selection, we assume that the SAM of helices are made of an equal and infinite number of left- and right-handed enantiomers, from which we randomly select domains contain n = 10 helices, the probability($P$) that these $n$ helices contain m right-handed enantiomer satisfies the formula,

$$P = C_n^m \left(\frac{1}{2}\right)^n \tag{12}$$

$$C_n^m = \frac{n!}{m!(n-m)!} \tag{13}$$

We define the chiral selectivity as $\sigma\% = \frac{m-(n-m)}{n} \times 100\%$, which produces the plot of Fig. 2e.

### Statistics and reproducibility
No statistical method was used to predetermine the sample size. Some CDS spectra showing drastically different features were not counted for the analyses as they are likely from nonspherical NPoMs. The particles were randomly collected and the investigators were blinded to allocation during experiments and outcome assessment.

## Data availability
All the data generated in this study are provided in the Supplementary Information/Source Data file. The NMR data are available from figshare with the DOI link https://doi.org/10.6084/m9.figshare.24249760. Source data are provided with this paper.

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

## Acknowledgements

This research is supported by the National Natural Science Foundation of China (12374356 T.D., 12204362 H.H., 12250410256 A.M.), the National Key Research and Development Program of China (2020YFA0211300 T.D., 2019YFB2203400 Z.W.), and the Fundamental Research Funds for the Central Universities (2042023kf0211 T.D., 2042023kf0233 Z.L./T.D.). We thank the helpful discussions with Profs. Duanduan Wan and Xuewen Chen and the assistance from Mengjun Xu for quartz crystal balance measurement. We also thank the Center for Nanoscience and Nanotechnology at Wuhan University for SEM characterizations.

## Author contributions

T.D. conceived the idea and designed the experiments. C.M., D.L., and Q.G. synthesized the helical supramolecules and performed the structural characterizations. C.Z. and Y.L. performed the spectra measurement and data analysis. H.H., C.Z., and X.W. performed the modeling and simulations. A.M., Z.W., and A.G. contributed to the verification of the modeling and revision of the manuscript. T.D. and Q.G. supervised the entire study. All the authors contribute to the writing and approve the final version of the manuscript. C.Z., H.H., and C.M. contributed equally to this work.

## Competing interests

The authors declare no competing interests.
