## [Peer Review File · Nature Communications]

Quantum plasmonics pushes chiral sensing limit to single molecules: a paradigm for chiral biodetectionsREVIEWER COMMENTS

Reviewer #1 (Remarks to the Author):

The manuscript describes the detection of chiral molecules using "cavities" formed between the a Au surface and nanoparticles. The authors demonstrate that quantum effects can amplify the sensitivity of the chiral detection. The observation of these "Quantum Plasmonic" phenomenon is interesting. However, I cannot recommend the manuscript for publication in its current forms because the claim of "single molecule" detection is not justified.

The definition of single molecule measurements would be, by consensus, the detection of single molecule in isolation. This is not what is reported by the authors. Indeed, from what I can gather from the paper, the authors are not even looking at a single nanoparticle (i.e what was achieved by the Baumberg group). Supplementary figure 34, clearly illustrates that within the field view of the microscope at least 10 particles would be observed. This number is insufficient to provide a true ensemble result, and thus why you get the data in figure 3 d and e. However, this is not a single particle / molecule measurement. In addition, within the cavity formed between the nanoparticle and the Au surface there are at least four molecules. However, the experiment do demonstrate that for the small sample size and racemic mixtures, an excesses of a single molecules can be detected.

So, to summarise my opinion, the authors focus on single molecule detection is not justified by the, albeit, interesting results they present. Although the method is sound the experiment is not sophisticated enough, (it would require methods based on scanning probe experiments and a method of to isolate a single molecule in the cavity) to achieve single molecule level measurements. With this being said I do believe that the quantum effects reported have the required novelty for publication in Nature Communication. I would be happy to review a revised manuscript which was focused on the quantum effects, rather than the spurious focus on single molecule detection.

Reviewer #2 (Remarks to the Author):

This work describes the detection of chiral molecules by plasmon coupled circular dichroism using a metal nanoparticle on a mirror geometry, in which the chiral analyte is sandwiched between the nanoparticle and the metal film. The chiroptical signal is read out out by single-particle dark-field scattering spectroscopy. The authors furthermore change the thickness of the analyte molecules that also serve as a spacer between the nanoparticles and the metal film. By approaching smaller distances and changing from a capacitive to conductive (quantum) coupling regime they find that while the electric field decreases in the quantum regime, the intensity of the circular dichroism actually increases again. Overall, this work is certainly novel and the results are interesting. I recommend that it is considered for publication in Nature Communications. However, the authors need to address a few important issues first in a revised manuscript.

1) The authors claim that they measure the chirality from single molecules. This is NOT true. While they have a limited number of molecules in the nanogap they fail to show that they have only one molecule. They are correct that they can detect an enantiomeric excess, but that is not what is typically accepted as a single-molecule experiment. This over-interpretation needs to be removed throughout the text or the authors need to demonstrate that indeed they measure the signal from only a single molecule.

2) I am concerned about the authors' claim that asymmetric excitation gives the same results as symmetric excitation (Figure S41). While the spectral shape might be maintained, the signal appears to be twice as large as the symmetric case. Considering that it is known that asymmetric excitation can cause extrinsic chirality, it is not OK to proceed with the asymmetric illumination, especially if also trying to rule out contributions from non-spherical nanoparticles and hence chirality of the substrate (see next comment).

3) One of the biggest challenges to assign plasmon coupled circular dichroism based on the chiral nature of the analyte is to rule out structural effects from the nanoparticle substrate itself. Overall, the authors do a good job by demonstrating opposite signs for the two enantiomers and the control experiment shown in Figure S33. However, it would be worthwhile in Figure S33 to not just give what appears like the absolute sign of the signal but also to specify if both positive and negative (20%) values were seen. Then, could these cases explain why the distributions shown in Figure 2c span all values for both enantiomers even though the main peak certainly switches upon changing to the other enantiomer? If so, I think it would be more appropriate to acknowledge that chirality of the substrate cannot be excluded in these colloidal systems even if using 'spherical' nanoparticles help to minimize these unwanted effects.

In general, the work is nice enough that it doesn't need over-interpretation (single-molecule detection and no complications from not perfectly achiral substrate) to be impactful. In fact, it would be better for the community to understand the challenges of detecting chiral molecules with plasmon coupled circular dichroism.

Reviewer #3 (Remarks to the Author):

The manuscript NCOMMS-23-14229 has demonstrated a quantum plasmonic paradigm for chiral bio-detection. The authors have addressed an important topic and their results are interesting. But their statements and their contribution to the research area are vague. I cannot recommend publication in the current form.

(1) In the Abstract, they described that the main problems hindering the achievement of single molecule sensitivity are the emergence of quantum plasmonic effects and the appearance of unintentional, non-biological plasmonic chirality (see lines 23-25). However, in the Introduction, they described that the main problems are the limited control on the gap size and orientation of the biomolecules therein (see lines 50-52). The author should make it clear which problems are solved in their experiments. Then, one can clearly see the contribution of this manuscript to the research area.

(2) In lines 58-63, the author tried to reveal the problem in the quantum region. But it is not clear what is the motivation for chiral sensing in the quantum region. In other words, readers of the manuscript would wonder what are the advantages of the quantum region over the classical region. The authors should address this issue clearly.

(3) In lines 77-88, they described that 'We reveal additional Coulomb interactions between the chiral supramolecules and tunnelling electrons compensate the decrease of local E-field incurred by tunnelling, which induces dramatic increase of CDSE in the quantum regime. This generalized theory of plasmon-enhanced chirality not only reveals the full picture of light-chiral matter interaction at different scales but also serves as a basic guideline for the design of advanced sensing platform for chiral molecules in general.' As I can see, they compared the experimental results with the predictions of the Quantum Corrected model. It is clear that tunneling electrons play important roles. On the one hand, tunneling electrons reduce the local E-field. On the other hand, tunneling electrons can also contribute to the absorption as indicated in lines 203-207. But, how the Coulomb interactions between the chiral supramolecules and tunnelling electrons increase CDSE is not clear. I guess that the chirality of supramolecules may make the tunneling itself highly enantioselective due to the Coulomb interactions, enhancing the enantioselectivities in the final signals. In addition, it is not clear what is 'this generalized theory'. If they meant the Quantum Corrected model, they should cite the reference. Otherwise, it is not suited to use the statement because they had not given a specialized theory to address how the Coulomb interactions between the chiral supramolecules and tunnelling electrons increase CDSE.

(4) They have used the g-factor as an important criterion to claim the single-molecule sensitivity. But, the g-factor was undefined in the manuscript. They should define the g-factor and explain why 'a g-factor up to 0.3' means the single-molecule sensitivity.

(5) Superchiral fields can qualify the CD enhancement of chiral molecules without a fixed orientation. In real experiments, the chiral molecules should have orientations. Hence, the enhancement of superchiral fields cannot identify accurately the CD signal. Authors should perform CD simulations with the consideration of chiral molecules with orientations (J. Phys. Chem. C 2017, 121, 666-675).

By the way, there are typos. The 'emergency' in line 23 should be replaced with 'emergence'. The 'region' in line 28 should be replaced with 'regions'.

Response Letter

We thank very much the Referees for their overall very supporting responses and several important and interesting comments. Therefore, we worked further on the presentation and data, incorporating the Referees' comments and making our presentation more transparent and precise. All changes are marked in yellow in the main manuscript. Below are detailed responses.

Reviewer #1:

(1) The definition of single molecule measurements would be, by consensus, the detection of single molecule in isolation. This is not what is reported by the authors. Indeed, from what I can gather from the paper, the authors are not even looking at a single nanoparticle (i.e. what was achieved by the Baumberg group). Supplementary figure 34, clearly illustrates that within the field view of the microscope at least 10 particles would be observed. This number is insufficient to provide a true ensemble result, and thus why you get the data in figure 3 d and e. However, this is not a single particle / molecule measurement. In addition, within the cavity formed between the nanoparticle and the Au surface there are at least four molecules. However, the experiment do demonstrate that for the small sample size and racemic mixtures, an excesses of a single molecules can be detected.

Thank you for your suggestion but, sorry, this is a strong misunderstanding. Now we have worked on the presentation to make all major points very transparent.

First, we are indeed looking at **the single particles** as the scattering signals were collected through a 100x objective with an optical fiber aperture of $\sim 50 \mu\text{m}$ (please see experimental details, actually the same setup as Baumberg group). The SEM images shown in Supplementary Figure 34 only suggest the Au NPs are well separated and the spectra we were collecting are indeed based on single Au NPoM structures with no interparticle coupling. Therefore, the data in Figure 3d and e are not correlated to this SEM image but rather over the whole samples with particle number over 50.

Second, what we mean by saying a single molecule in our text is our method's capability of detecting single molecule according to the signal-to-noise ratio for current measurement but not really we are measuring single chiral molecules as at least 4 helices are placed within the gap if dense layer of SAM was formed according to calculation. To avoid this misunderstanding, we now adjust the statement of single molecule detection as *"it holds potential capability for single chiral molecule detection."* In addition, the title emphasized the single-molecule detection capability as a potential method for the next future, "... pushes chiral sensing limit to single molecules ...", - the word "pushing" means that we show the direction of development in the field of chiral sensing, but we did not achieve that limit yet.

In addition, thanks for the excellent comment! - “However, the experiment do demonstrate that for the small sample size and racemic mixtures, an excesses of a single molecules can be detected.” In the abstract, we now add a phrase: “*The minimum number of detected molecules based on single-Au particle is about four, which allows for the detection of an enantiomeric excess of a sample with potential sensing capability of single molecule.*”

Reviewer #2:

- (1) The authors claim that they measure the chirality from single molecules. This is NOT true. While they have a limited number of molecules in the nanogap they fail to show that they have only one molecule. They are correct that they can detect an enantiomeric excess, but that is not what is typically accepted as a single-molecule experiment. This over-interpretation needs to be removed throughout the text or the authors need to demonstrate that indeed they measure the signal from only a single molecule.

Thank you much for the important suggestion that should remove any potential confusion of the reader! We fully agree that we are not measuring single chiral molecules but probably a few chiral molecules (~4-10) according to the calculation. Yet, importantly, we do single-particle spectroscopy in the sense of AuNP-mirror gap antennas! Now in the text we clarify these important properties that are the keys for our chiral detection method.

The number of molecules detected is largely limited by the sample preparation but from the signal-to-noise ratio, single chiral molecule detection is possible with this strategy as we interpolate the CDS intensity to one molecule. This is important for the potential development of this field, which possesses lots of challenges. To be more rigorous, we adjusted the statement as “*potential capability of single molecule detection*”. Also, we added several new lines to emphasizes “single NP experiment”, “a few molecule chiral detection”, “potential chiral single-molecule detection”, “*the detection of an enantiomeric excess of a sample (since we see a few molecules using a single AuNP gap-antenna*”.

Here are the new texts in the manuscript:

Page 7 :

NPoM-assisted chiral sensing of a few chiral helices.

Thus, it holds great potential for chiral single-molecule detection.

Page 8 :

Another characteristic of single chiral molecule sensitivity is that the racemic mixture that appears achiral as entity can be locally resolved as enantiomer excess (ee) due to the population fluctuation in the single Au NPoM gap.

- (2) I am concerned about the authors' claim that asymmetric excitation gives the same results as symmetric excitation (Figure S41). While the spectral shape might be maintained, the signal appears to be twice as large as the symmetric case. Considering that it is known that asymmetric excitation can cause extrinsic chirality, it is not OK to proceed with the asymmetric illumination, especially if also trying to rule out contributions from non-spherical nanoparticles and hence chirality of the substrate (see next comment).

Another important comment - thank you. Firstly we have to clarify that the so-called asymmetric excitation is due to the artefact of our microscope that the illumination light

is not perfect spherical as some parts are slightly blocked in the beam path by the frame of the waveplate holder. We noted this artefact and tried to avoid the blocking to get perfect symmetric excitation and measured the CDS overall over again. The statistic results are summarized in Fig. R1b. As compared to results of the asymmetric incidence (Fig. R1a, adapted from Fig. 2c), the results from symmetric incidence show the same distribution span from negative to positive but with much weaker intensity (mostly reduced to zero). It is true as the Referee suggests that the substrate effect indeed may cause and induce some interruption on the chiroptic measurement. But the dominant CDS values always reflect the chirality of the molecules. We have now added the results of symmetric incidence in SM Fig. 43 and clarify the issue mentioned by the referee in the manuscript at the bottom of page 12.

Fig. R1 Statistic occurrence of CDS peak intensity based on 50 particles for Au NPoM with P and M-helices in the gap (a) asymmetric incidence showing CDS with average intensity of $\sim 20\%$ and (b) symmetric incidence showing CDS with average intensity of $\sim 10\%$.

- (3) One of the biggest challenges to assign plasmon coupled circular dichroism based on the chiral nature of the analyte is to rule out structural effects from the nanoparticle substrate itself. Overall, the authors do a good job by demonstrating opposite signs for the two enantiomers and the control experiment shown in Figure S33. However, it would be worthwhile in Figure S33 to not just give what appears like the absolute sign of the signal but also to specify if both positive and negative (20%) values were seen. Then, could these cases explain why the distributions shown in Figure 2c span all values for both enantiomers even though the main peak certainly switches upon changing to the other enantiomer? If so, I think it would be more appropriate to acknowledge that chirality of the substrate cannot be excluded in these colloidal systems even if using 'spherical' nanoparticles help to minimize these unwanted effects.

This is a valid suggestion and we now plot the distribution of both positive and negative values in the figure (Fig. R2, also updated in Fig. S33), which indeed show the same distribution as Fig 2c. And we agree that the substrate effect indeed cannot be excluded. This is clarified in the manuscript.

Fig. R2 Statistic occurrence of CDS peak based on 50 particles for Au NPoM with BDT in the gap.

Reviewer #3:

(1) In the Abstract, they described that the main problems hindering the achievement of single molecule sensitivity are the emergence of quantum plasmonic effects and the appearance of unintentional, non-biological plasmonic chirality (see lines 23-25). However, in the Introduction, they described that the main problems are the limited control on the gap size and orientation of the biomolecules therein (see lines 50-52). The author should make it clear which problems are solved in their experiments. Then, one can clearly see the contribution of this manuscript to the research area.

Thank you much for the comment and observation. Both of the problems are the current issues of chiral sensing system and in our self-assembled plasmonic system, we solved both problem of quantum plasmonic enhancement and the orientation of chiral molecules. But perhaps the former is a major problem and the latter is minor. In the abstract, we only focus on major problem and in the introduction, we actually point out both major (lines 60-63: "... However, this might not always be true for optical chirality of a tunnelling structure, which is ambiguously understood.") and minor (lines 50-52: "However, due to the limited control on the gap size and orientation of the biomolecules therein, ...") problems. We hope this clarifies the concern of this referee. We also highlight the problems we solved in this area in the discussion section: "...the accessibility of quantum plasmonic region with the alignment of chiral dipoles to the plasmon dipole, which reveals additional CDSE from the quantum tunnelling effect, unveiling the full description of chiral light-matter interactions."

(2) In lines 58-63, the author tried to reveal the problem in the quantum region. But it is not clear what is the motivation for chiral sensing in the quantum region. In other words, readers of the manuscript would wonder what are the advantages of the quantum region over the classical region. The authors should address this issue clearly.

Thank you for your thoughts. People always aim to extend the limit of plasmonic enhancement. In the classical regime, the plasmonic enhancement is not large due to its large gap size while in quantum regime, although the mode volume is much smaller, the enhancement for SERS may be deteriorated due to electron tunneling. However, what we found quantum plasmonics is actually good for extending chiral enhancement limit. This is clearly beneficial for higher chiral sensitivity with the possibility of single molecule detection. This is further clarified in the introduction via the following lines: "This quantum tunnelling effect is commonly recognized to be detrimental to the plasmonic enhancement due to the reduced local E-field in the gaps. However, this might not always be true for optical chirality of a tunnelling structure, which is ambiguously understood".

Why so? Future theoretical computations should explain this important observation, motivating the field further. Qualitatively, this is because the SERS happens on one molecular site and the EM enhancement is crucial, while the chirality of an electronic orbital is more extended and it is a weaker effect optically, and therefore, to achieve the needed sensitivity, we need to go to the quantum regime anyway.

- (3) In lines 77-88, they described that ‘We reveal additional Coulomb interactions between the chiral supramolecules and tunnelling electrons compensate the decrease of local E-field incurred by tunnelling, which induces dramatic increase of CDSE in the quantum regime. This generalized theory of plasmon-enhanced chirality not only reveals the full picture of light-chiral matter interaction at different scales but also serves as a basic guideline for the design of advanced sensing platform for chiral molecules in general.’ As I can see, they compared the experimental results with the predictions of the Quantum Corrected model. It is clear that tunneling electrons play important roles. On the one hand, tunneling electrons reduce the local E-field. On the other hand, tunneling electrons can also contribute to the absorption as indicated in lines 203-207. But, how the Coulomb interactions between the chiral supramolecules and tunnelling electrons increase CDSE is not clear. I guess that the chirality of supramolecules may make the tunneling itself highly enantioselective due to the Coulomb interactions, enhancing the enantioselectivities in the final signals. In addition, it is not clear what is ‘this generalized theory’. If they meant the Quantum Corrected model, they should cite the reference. Otherwise, it is not suited to use the statement because they had not given a specialized theory to address how the Coulomb interactions between the chiral supramolecules and tunnelling electrons increase CDSE.

Yes, this is indeed the mechanism we are proposing: the chiral molecules induce the tunneling electrons enantioselectively via Coulomb interactions, which results in larger difference of the chiral scattering. As for “generalized theory”, we mean the explanation of chiral transfer via coulomb interactions in both classical and quantum region. Perhaps it is not suitable to use “theory” here, hence we change this term to “model”.

- (4) They have used the g-factor as an important criterion to claim the single-molecule sensitivity. But, the g-factor was undefined in the manuscript. They should define the g-factor and explain why ‘a g-factor up to 0.3’ means the single-molecule sensitivity.

Thank you – indeed it should be explicitly defined. The g-factor is defined as

$$g = \frac{2(S_{LCP} - S_{RCP})}{(S_{LCP} + S_{RCP})}$$

S_{LCP} and S_{RCP} are scattering spectra of both incidences. This is the same as CDS spectra, which is now clarified in the Methods section.

As for single molecule sensitivity, we calculate the number of helical oligomers in the gap which is around 10 (see the discussion in the last paragraph of Page 7). Suppose the CDS intensity and the number of helices correlate linearly, following the Lambert-Beer law, we can interpolate the intensity of one helical molecule, which is $0.2/10=0.02$ (2%). This value is on the margin of signal-to-noise ratio of the detection. Thus, it is reasonable to claim it has the potential for single-molecule sensitivity.

- (5) Superchiral fields can qualify the CD enhancement of chiral molecules without a fixed orientation. In real experiments, the chiral molecules should have orientations. Hence, the enhancement of superchiral fields cannot identify accurately the CD signal. Authors

should perform CD simulations with the consideration of chiral molecules with orientations (J. Phys. Chem. C 2017, 121, 666-675).

We agree the molecular orientation may contribute additional enhancement and now performed the CDS simulation considering the molecule orientation according to the JPCC paper by Prof. Zhang. We first reproduced Fig. 1a of the JPCC paper in Fig. R3a to make sure our model is correct and valid. We then applied this model to NPoM structure with oriented molecules in the gap as shown in the schematic inset of Fig 2a (the angle between electric and magnetic dipole is roughly 20 degrees). The results shown in Fig. R3b suggest the trend considering molecule orientation (anisotropic) is the same as our previous simulation in the manuscript (isotropic) but seems to have a larger CD enhancement factor. Clearly, the consideration of molecular orientation does not disturb the enhancement contributed by quantum plasmonics as they all show the same trend of nonlinear increase in the quantum tunneling regime. We provide this data in supplementary materials and related discussion is added in the manuscript: please see page 10, last a few lines: *“The overall trend of enhancement is not disturbed by the orientation of the molecules in the gap although the alignment of the molecular dipole to the plasmonic dipole can further increase the CDSE by 3-fold (Supplementary Fig. 41).”*

Fig. R3 CDS simulation considering the molecule orientation. (a) Reproduced results of the JPCC paper (b) Comparison of CDSE calculated with and without the consideration of molecular orientation

- (6) The 'emergency' in line 23 should be replaced with 'emergence'. The 'region' in line 28 should be replaced with 'regions'.

Thank you for pointing out the typos and they are now corrected.

REVIEWERS' COMMENTS

Reviewer #1 (Remarks to the Author):

I'm happy that the authors have addressed my comments

Reviewer #2 (Remarks to the Author):

The authors have adequately addressed my prior concerns. The manuscript can be accepted.

Reviewer #3 (Remarks to the Author):

I think the questions are well-addressed and the manuscript can be published.